# *Generalized models to estimate carbon and nitrogen stocks of organic soil horizons in Interior Alaska*

Kristen Manies, Mark Waldrop, Jennifer Harden

U.S. Geological Survey
345 Middlefield Rd.
Menlo Park, CA 94025 USA

*Correspondence to*: Kristen Manies (kmanies@usgs.gov)

**Abstract**
Boreal ecosystems comprise one tenth of the world's land surface and contain over 20 % of the
global soil carbon (C) stocks. Boreal soils are unique in that its mineral soil is covered by what can be
quite thick layers of organic soil. These organic soil layers, or horizons, can differ in their state of
decomposition, source vegetation, and disturbance history. These differences result in varying soil
properties (bulk density, C concentration, and nitrogen (N) concentration) among soil horizons. Here we
summarize these soil properties, as represented by over 3000 samples from Interior Alaska, and examine
how soil drainage and stand age affect these attributes. The summary values presented here can be used to
gap-fill large datasets when important soil properties were not measured, provide data to initialize
process-based models, and validate model results. These data are available at
https://doi.org/10.5066/P960N1F9 (Manies, 2019).
**1 Introduction**
Boreal soils play an important role in the global carbon (C) budget and are estimated to store
between 375 - 690 Pg of C (Hugelius et al., 2014; Bradshaw and Warkentin, 2015; Khvorostyanov et al.,
2008), which is over 20 % of the global soil C stock (Jackson et al., 2017). A large portion of this C can
be found within the organic soil layer (Jorgenson et al., 2013). Although plant inputs into the soil can be
relatively high during the summer, C losses from the soil are low, as cool and/or freezing soil
temperatures result in low rates of decomposition. The imbalance between C inputs and losses results in
organic soils that can be quite thick and store large amounts of C (Jorgenson et al., 2013). There is also
considerable C found in the mineral soil of these systems, especially where protected by permafrost
(O'Donnell et al., 2011). Thus, both organic and mineral soil play an important role determining the
amount of C stored in boreal ecosystems.
Nitrogen (N) also plays an important role in boreal ecosystems due to N limitations on plant
growth (Herndon et al., 2020). N inputs to boreal ecosystems often begin with N fixation from
cyanobacteria, usually associated with mosses, or symbiotic actinomycetes, mainly the genus *Frankia*.
Net N mineralization increases over the course of upland succession, until the oldest state, black spruce
(*Picea mariana*) forest, when rates drop sharply (Kielland et al., 2006). Boreal ecosystems can have N
restricted by certain species, such as *Sphagnum* spp., through competitive interactions and slow rates of
turnover (Malmer et al., 2003). In addition, N cycling can become limited due to environmental factors
such as permafrost or anerobic conditions (Limpens et al., 2006; Bonan, 1990). Once released, N
availability impacts decomposition and plant growth and, therefore, can also influence rates of C
accumulation and loss.
Boreal organic soils are unique when compared to soils from other regions. These organic soils
can be thick, ranging from several centimeters to several meters (Ping et al., 2006). They are also
comprised of layers, or horizons, which as they deepen and increase in age also increase in their degree of
decomposition. These organic soil horizons are also influenced by the vegetation from which they formed
(Deluca and Boisvenue, 2012). Vegetative history is usually determined by post-disturbance plant
succession. Age and vegetative history not only affect the soil density, but also C and N concentrations,
resulting in large differences in C and N storage among horizons.
The main disturbances that affect boreal soil properties are fire and permafrost thaw. Fires affect
boreal soils through the combustion of litter and surface organic layers (as ground fuel; Harden et al.,
2000), with the amount and depth of combustion regulated by fire severity (Turetsky et al., 2011). Fire
directly effects surface organic soils, both in elemental composition and structure (Neff et al., 2005). In
addition, there are indirect effects of fire on soil properties. The loss of insulating organic soil results in a
darkened soil surface, which in turn warms post-fire soils, increasing decomposition rates from the
surface downward (Genet et al., 2013; O'Neill et al., 2002). In addition, both fire return interval and fire
severity influence post-fire vegetation and the re-accumulation of organic soil layers. As different tree and
understory species have different amounts of C and N in their tissues (Van Cleve et al., 1983), changes in
post-fire vegetation affect soil C and N accumulation rates and thus, the concentration of these elements
in surface soil. Permafrost thaw also affects soil properties in several ways. By definition, thaw exposes
older, previously sequestered C to warmer soil temperatures (Osterkamp et al., 2009), increasing rates of
decomposition (Mu et al., 2016; Schadel et al., 2016). In well drained sites post-thaw conditions usually
result in water draining from the soil, resulting in oxic conditions (Estop-Aragonés et al., 2018). In
lowlands, permafrost thaw often results in subsidence and inundation, changing the ecosystem from a
forested permafrost plateau to a thermokarst wetland (Schuur et al., 2015). Fire can often be a trigger for
this rapid permafrost thaw (Myers-Smith et al., 2008). Post fire vegetation changes affects both C and N
inputs, again affecting the concentration of these elements within surface organic soil layers. As both fire
frequency and permafrost thaw are expected to increase in the future (Hinzman et al., 2005),
biogeochemical models have a need to characterize how these disturbances will impact C and N stocks.
To accurately represent future scenarios, models need to include the distinct properties of organic soil
horizons found in the boreal region (Flato et al., 2013).

Despite the need to accurately portray the state and dynamic nature of boreal organic soil

properties, these soils have not been widely characterized nor compiled into a common framework.
Instead, much of the work regarding boreal soils has focused on predicting C and N stocks for combined
organic and mineral soil horizons to a predetermined depth (Johnson et al., 2011; Bauer et al., 2006). Ping
(2010) examined organic soils for Alaska, but only focused on black spruce (*Picea mariana*) forests. In
addition, Michaelson et al. (2013) compiled a great deal of Alaskan-based soil data, although they present
these data for the organic soil layer as a whole. Therefore, there is currently no source of summarized data
of soil properties by organic soil horizon. To fill this gap, we summarized soil properties from a database
of over 3000 observations from Interior Alaska (Figure 1). Soil properties were categorized by degree of
decomposition (via classification into distinct organic soil horizons), soil drainage, and stand age. This
data set can be used in many ways including field comparisons, models construction, and model
validation.


## 2 Methods

### 2.1 Field site classifications

Soil cores were sampled at 58 different sites located within several areas of Interior Alaska (Figure 1). Several different ecosystem types were sampled, including black spruce forests (~50%), wetlands (~26%), and deciduous and mixed forests (~16%). Between 1 and 14 soil profiles were sampled at each site, for a total of 292 soil profiles. Sampling took place over a 15-year period from 2000-2015. We examined the effect of fire or permafrost thaw disturbance on soil properties by categorizing each of the soil profiles in relation to time since the last disturbance, which we divided into three age classes: new (<5 yrs old), young (5 – 50 yrs old), and mature (> 50 yrs old). All new sites were recently burned and thus had lost some portion of their surface organic horizons (Harden et al., 2000), while young sites experienced either fire or permafrost thaw.

In addition, sites were classified according to their soil drainage. Although classifications of soil drainage have been established for many soil types (Soil Survey Division Staff, 1993), the presence of permafrost, and its effect on drainage and soil moisture, necessitates modifications of this system (Expert Committee on Soil Survey, 1982). Although generally described (Harden et al., 2003; Johnstone et al., 2008), a soil drainage classification for permafrost landscapes is lacking. Here we present a soil drainage classification decision tree, developed over the past two decades, for areas of discontinuous permafrost (Figure 2). Well drained sites are similar to traditional drainage classifications, in that water moves through the soil rapidly. However, moderately well drained drainage sites have permafrost between 75 – 150 cm, which increases soil moisture of surface organics. Somewhat poorly, poorly, and very poorly drained sites have some factor (permafrost, soil texture, or landscape position) that inhibits drainage and causes redoximorphic features such as blue-grey colors in the mineral soil to appear. Somewhat poorly drained sites have a shallow active layer (often around 50 cm), which affects soil moisture and surface vegetation. Poorly drained sites experience saturated surface conditions only while seasonal ice is present (usually May through early July), while very poorly drained sites have saturated surface soils during the entire growing season.

Modification of the drainage class occurs when sites are on a slope. When sites are located on a slope
of greater than 5 %, drainage increases (Woo, 1986; Carey and Woo, 1999), and therefore drainage class
designation (Figure 2) is increased by one step. This is called the hillslope modifier. In addition, because
burning increases active layer thickness (Gibson et al., 2018), recently burned sites may have deeper
permafrost or no permafrost at all. Because the effects of these drier soil properties may not have yet
propagated through factors such as thickness of the deeper organic layers, for many analyses, including this
paper, it makes more sense to ascribe their soil drainage using nearby unburned sites.

**2.2 Soil sampling methodology**
Soil cores were obtained using several different methods.  The first method, most often used with
surface horizons, involved cutting soil blocks to a known volume. Another method often used inovlves a
coring device inserted into a hand drill (4.8 cm diameter; Nalder and Wein, 1998). Wetter sites were
sometimes sampled while frozen using a Snow, Ice, and Permafrost Research Establishment (SIPRE)
corer (7.6-cm diameter; Rand and Mellor, 1985). Alternatively, if wetter sites were sampled unfrozen we
used a 'frozen finger'. This coring method uses a thin-walled, hollow tube (~6.5 cm diameter), sealed at
one end, which is inserted into the ground until it hits mineral soil. A slurry of dry ice and alcohol is then
poured into the corer, freezing the unfrozen material surrounding the corer to the outside. The corer is
removed and the exterior of the core is scraped to remove any large roots or material that stuck to the
sample during removal. Another method occasionally used in unfrozen saturated soils involves the
insertion and careful removal of PVC tubing sharpened on one end. Finally, a variety of commercially- or
home-made soil corers were used to obtain volumetric samples for ~6% of these data, usually for mineral
soil samples. For some soil profiles, two coring methods were combined to create continuous samples
from the surface to the mineral soil. While most cores were sampled into the mineral soil, some cores
ended at or before the organic/mineral interface due to the presence of permafrost without proper
sampling equipment or because the cores were collected for the purpose of only studying surface
organics. All sampling methods were volumetric, providing the basis for bulk density calculations (g/cm$^3$)
Organic soil layers or horizons were described and then subdivided according to field-based
visual and tactical factors such as level of decomposition, color, and root abundance, regardless of region
or soil drainage. These horizons provided the basis for our analyses and are based on Canadian (Soil
Classification Working Group, 1998) and U.S. Department of Agriculture's Natural Resource
Conservation Service (Soil Survey Staff, 1998) soil survey techniques. A description of the horizons and
the codes we used to represent them are found in Table 1, but in summary there are six main horizons:
live moss (L), dead moss (D), fibric (mostly undecomposed; F), mesic (more decomposed; M), humic
(very decomposed; H), and mineral soil (Min).
To aid researchers who may need to have these properties summarized in a more simplified scheme (as
in Yi et al., 2009; O'Donnell et al., 2009), we also combined horizons post-hoc into a simplified scheme.
Here, the fibrous horizon consists of both the dead moss (D) and fibric (F) horizons, while the amorphous
horizon combined the mesic (M) and humic (H) horizons. These combinations were based on similarities
in decomposition state and depth within the organic soil profile. We also present data for several types of
surface horizons that are only found a small fraction of sites; those data are presented separately. Ash and
burned organic surface horizons are only found in recently burned sites.  Lichen and litter dominated
horizons are only found on the surface of ~16 % of profiles and related to well drained forest conditions.
Our field studies also found several horizon types (buried wood, grass, etc.) for which we had few
observations (5 or less), and, thus, were not included in our analyses.

**2.3 Laboratory methodology**
Once returned from the field soils horizon samples were weighed and air-dried at room temperature
(20 °C to 30 °C) to a constant mass, then oven-dried for 24-48 hours in a forced-draft oven. Organic soils
(live moss, dead moss, fibric, mesic, and humic horizons) were oven-dried at 65 °C to avoid the alteration
of organic matter chemistry. Mineral soils were oven-dried at 105 °C. Mineral soil samples were gently
crushed using a mortar and pestle, with care to break only aggregates, and then sieved through a 2-mm
screen. Soil particles that did not pass through the screen were removed, weighed, and saved separately;
soil that passed through the screen was then ground by using a mortar and pestle to pass through a 60-mesh
(0.246-mm) screen. The ground material was mixed and placed in a labeled glass sample bottle for
subsequent analyses. Organic soil samples were weighed, and roots wider than 1 cm in diameter were
removed, weighed, and saved separately. The remaining sample material was then milled in an Udy Corp.
Cyclone Sample Mill to pass through a 0.25-mm screen and placed in a labeled glass vial.

We analyzed soil samples for total C and N using a Carlo Erba NA1500 elemental analyzer

(Fisons Instruments). Samples were combusted in the presence of excess oxygen. The resulting sample
gases were carried by a continuous flow of helium through an oxidation furnace, followed by a reduction
furnace, to yield $CO_2$, $N_2$, and water vapor. Water was removed by a chemical trap and $CO_2$ and $N_2$ were
chromatographically separated before the quantification of C and N (Pella, 1990a,b). We assumed that
mineral soil samples below pH 7, which are common to Interior AK, had no inorganic carbon (IC)
present, and thus total C represents total organic C. For mineral-soil horizons were IC was present, we
removed carbonates using the acid fumigation technique (Komada et al., 2008) prior to running samples.
To do this, we preweighed samples in silver capsules and transferred them to a desiccator. Samples were
wetted with 50 μL of deionized water and then exposed to vaporous hydrochloric acid (1 N) for a
minimum of 6 hours, during which carbonates degassed from samples as carbon dioxide.

**2.4 Data quality and statistical methodology**

Often the soil descriptions at the interface of the organic and mineral soil included notations

indicating that these horizons consisted of mixed organics and mineral soil. Using visual and textural cues
the field, horizons were categorized as either mineral (< 20 % C) or organic (≥ 20 % C). However,
chemistry data sometimes shows these horizons were miscategorized due to slight under or over
estimations of OM content (for example, a mineral soil with 22 % C). We used C chemistry to remove
organic soils with < 20 % C from our analyses.

All statistical analyses were run using the R program (R Core Team, 2017). Data were

transformed to meet assumptions of normality (Table S1).  The effects of drainage and age class for all
soil horizons with the exception of the fibrous and amorphous horizons, was tested for significant
difference among the different soil horizons using the mixed-effects model command *lmer* (lme4; Bates et
al., 2015), using soil profile (or soil core) as the random effect. When significant, differences among
drainage types or age class were determined using estimated marginal means (Least-squares means;
*emmeans*) (Lenth et al., 2020). No interactions were examined. Evaluation for the fibrous and amorphous
horizons, because all samples were within a single soil profile, was done using the analysis of variance
model (*aov*) with the Tukey honestly significant difference (*TukeyHSD)* function.

**3. Dataset Review**
**3.1 Bulk density**

Bulk density varied by depth and was significantly different ($p < 0.05$) among all horizon types

(live moss, dead moss, fibric, mesic, humic, and mineral soil; Table 1). Surprisingly, as they are
comprised of very similar material, even the live and dead moss horizons had significantly different bulk
densities. Bulk density increases ~10-fold from one organic horizon to the next down the soil profile
(from 0.022 g cm$^{-3}$ for live moss to 0.215 g cm$^{-3}$ for the humic horizon). These differences are likely
related to the length of time each soil horizon has had to decompose. As soil horizons become older, plant
fibers break down physically and biologically, becoming smaller and more compressed.

Bulk density also varied by drainage class, particularly at the deeper depths. Well drained sites

tended to have higher bulk densities than other poorer soil drainage classes, especially for the deeper soil
horizons (e.g. fibric and mesic; Table S2).  Higher bulk densities with better drainage is likely related to
two factors: 1) the influence of lichens and litter, which often found at well drained sites, and have higher
bulk densities than moss (Table 4), and 2) the influence of mineral soil, which, due to shallower organic
soils, is more likely to be incorporated into fibric (F) and mesic (M) horizons. Greater mineral
incorporation into organic layers of shallow well drained soils is supported by the lower % C values also
found within well-drained F and M horizons (Table S3).  New (< 5 yr old) sites often had higher bulk
densities than the older age classes (Table S2). There were, however, very few significant differences in
bulk density by age class, so this factor does not appear to play strong role in determining bulk density.

**3.2 Carbon**
Upper, shallow organic soil horizons (live moss, dead moss, and fibric horizons) differ from
deeper horizons (mesic and humic horizons) in several respects. Shallow horizons are consistently higher
in % C than deeper horizons (Table 2). However, upper, shallow horizons are lower in bulk density than
deeper horizons (Table 2), so that C density values (g cm$^{-3}$) increase dramatically with depth (Figure 3).
Therefore, even though the deeper organic horizons (M and H) have slightly lower C concentrations than
the shallow horizons, their high bulk densities result in large amounts of C at depth. In fact, given average
thickness, bulk density, and % C (Table 2), approximately 75% of the soil C is stored in the mesic and
humic soil horizons.
There were few clear trends with C concentration with drainage class, although moderately well
drained sites usually had higher C concentrations than the other drainage classes, especially somewhat
poorly drained sites (Table S3). Lower C values for the fibric and mesic well-drained sites are likely due
to the inclusion of mineral soil material into these horizons. While this difference is likely due in large
part to natural process such as cryoturbation or aeolian contributions, these horizons are thinner in well
drained sites (Table 3), so any accidental inclusion of mineral soil within these horizons during sampling
would have more of an effect.
C concentration increased with increasing age class for all organic horizons but the humic horizon
(Table S3). Since all sites classified as 'new' were recently disturbed by fire, this increase could be due to
the inclusion of more live roots and/or the loss of ash in older stands. Ash has a lower C content (Table 4)
and is a component of recently burned soil's surface horizons.


**3.3 Nitrogen**


N concentration within the organic horizons increased with depth and then declined again in the
mineral soil (Table 2). There was significant variability in N by drainage class for each horizon type
(Table S4).  The poorly and very poorly drained sites had greater concentrations of N than other drainage
classes for the fibric (F), mesic (M), and humic (H), and mineral horizons, and lower concentrations of N
in the dead moss (D) horizon. These higher values are likely because N builds up under saturated
conditions, due to low rates of microbial activity, limiting decomposition (Limpens et al., 2006).  There
was also more N in the live and dead moss horizons of the new and younger stands (Table S3). These
differences are likely related to differences in N quality of early succession litterfall (Bonan, 1990).

**3.4 C:N ratio**


C:N ratios patterns followed those of C and N, with the surface organic horizons (live moss, dead
moss, and fibrics) having more similar values than the deeper soil organic horizons (Table 2). Well
drained sites tended to have lower C:N ratios (Table S5), likely caused by the lower C concentrations
found there (see section 3.2). C:N ratio increased with age class, but only in the surface organic horizons
(live moss, dead moss, and fibrics). These trends appear to be more influenced by differences in N by age
class than changes in C.

**3.5 Soil horizon thickness**


The factor that varied the most by horizon was the thickness of each horizon type (Table 2), and,
unlike most of the other factors, the standard deviation was often greater than the mean. There was a very
strong effect of drainage on horizon thickness, with the well-drained sites having much thinner soil
horizons (and no humic horizon) than the other drainage classes and the very poorly drained sites having
much thicker soil horizons that the other drainage classes (Table 3). Age class also plays a role in horizon
thickness: new sites (<5 yrs old) had much thinner organic soil horizons than young or mature sites (Table
3). Since new sites recently burned, these thin soil horizons are the result of the loss of organics due to
combustion. Both fire return interval and fire severity impact the amount of legacy soil remaining
(Harden et al., 2012), therefore fire history likely plays a large role in horizon thickness.

Vegetation could also influence horizon thickness. An examination of these data that included

current surface vegetation found greater thicknesses for sites with *Sphagnum* sp. and sedges, although this
factor usually was not statistically significant. Historical vegetation could also influence horizon
thickness. For instance, if a site was *Sphagnum* dominated in the past, even if it is not the current surface
vegetation, the soil profile is more likely to have thicker soil horizons due to the slow decomposition rate
of *Sphagnum* (Turetsky et al., 2008). Because such historical factors are difficult to measure and predict,
we recommend that researchers obtain their own measurements of organic horizon thickness whenever
possible and, if using the thickness data presented in Table 3, account for the variability found for
thickness estimates in their analyses.

**4.0 Discussion of the data set**
**4.1 Comparison to other data sets**

Our data are the first of its kind to present organic horizon data across a range of Alaskan boreal

ecosystems.  Other studies have examined organic soil as a separate entity from mineral soil but with
certain limitations. Michaelson et al. (2013) used Alaskan USDA-NRCS soil pedon data to examine soil
properties of both organic and mineral soil but present these data for the organic portion as a whole. This
study shows that there is significant variation in bulk density and C and N concentration across organic
horizons, and therefore, one should not disregard these horizon-based variations.  In a separate study,
Ping et al. (2010) separated the organics into two horizons from boreal black spruce stands ($O_{surface}$,
$O_e/O_a$). Our study supports the results of Ping et al., (2010), which found a decrease in C:N ratios with
increasing depth. Moreover, our study provides data from a fuller suite of soil horizons and includes data
from bogs, fens, and deciduous forests.


**4.2 How well do these values represent other data?**

We tested how well our data from Interior AK can predict C and N stocks in other studies. Our first test was for 142 samples taken from two fire chronosequences located near Thompson, Manitoba (Manies et al., 2006). Each chronosequence represents a different drainage class: moderately well drained versus somewhat poorly drained. These data were based on the same methods of sampling and describing soil horizons. Using the horizon designations (Table 1) and horizon thickness (cm) from the Canadian data, we assigned bulk density, C, and N values (Table 2). These predicted horizon-based C and N stocks were summed for each soil profile and compared to the measured values. We found our predicted stocks were relatively evenly distributed between being lower or higher than measured stocks (Figure S1), with the majority of estimated stocks (>85%) within 50% of measured stocks and over 60% within 20% of measured stocks. Soil profiles with much higher predicted than measured stocks were due to very low measured bulk densities (e.g., a measured bulk density for a fibric horizon of 0.01 $g/cm^3$, as compared to the predicted value of 0.06 $g/cm^3$). The differences we found between measured and predicted stocks could be due to regional differences between the Alaskan and Canadian sites in factors, such as disturbance history or vegetation composition. In addition, accurate bulk density measurements is time consuming to do correctly (Nalder and Wein, 1998) and could also play a role.

To further explore the predictive capabilities of our data, we also compared predicted versus measured C stocks for a second study, this one located within Alaska (Kane and Ping, 2004), in which horizon thickness (all samples), % C (all samples), and bulk density (one 5.08 cm diameter sample per horizon per site) for soil profiles were measured along a continuum of tree productivity. To calculate predicted C stocks we used their thickness values with bulk density and % C values from Table. 1. However, Kane and Ping (2004) used the US Soil System to describe and sample their soils, dividing the organic soil profile into $O_i$ and $O_e/O_a$ horizons. We chose to represent their $O_i$ data, which they described as slightly decomposed moss, using our fibrous horizon and their $O_e/O_a$ data, which they described as intermediately decomposed moss with rare saprics, as our amorphous horizon. Predicted C stocks were higher than measured stocks (Figure S2). This result was mostly due to differences in bulk density values

between our amorphous horizon and their $O_e/O_a$ horizon. Their study had $O_e/O_a$ bulk density values that
ranged between 0.06 and 0.12 $g/cm^2$, which is typical of our fibric (F) and mesic (M) horizons (Table 2).
When we model their $O_e/O_a$ data using F values, we slightly underestimate stocks, while if we model their
$O_e/O_a$ data using M values we slightly overestimate their stocks (Figure S2). Thus, bulk density
measurements play a role in these differences. These results also demonstrate that soil description
protocols play an important role in characterizing C and N stocks and, in this case, the different system
used to identify and sample organic soil horizons may not be equivalent.

**4.3 Caveats and suggestions for use**
One of the important uses of this dataset is the potential for estimating C and N stocks based on
simple field characterizations of organic soil horizons of North American boreal forests and wetlands.
Because soil sampling and processing is quite time intensive, researchers may decide to measure
thicknesses of the various soil horizons within their sites, using the descriptors in Table 1, and then
calculate C and N stocks using the average values presented in Tables 2, S2, S3, or S4. This approach
minimizes errors associated with the high variability found for horizon thicknesses, due to variable site
histories.
While C stocks of mineral soils were not evaluated in this study, this region contains large
amounts of C within mineral soils, especially within Yedoma deposits (Hugelius et al., 2014; O'Donnell
et al., 2011). The mineral soil data presented here represent mostly the uppermost mineral soil. Additional
examinations into bulk density and C concentrations of Alaskan mineral soil can be found in Ping et al.
(2010), Michaelson et al. (2013), and Ebel et al. (2019).
Although our data provide an important resource for several properties of organic horizons, we
acknowledge that our samples are dominated by mature sites from areas that are not well drained.
Therefore, as additional soil horizon data is sampled, we encourage researchers to expand upon the work
presented here.

**5 Data Access**
All data used in this manuscript are available from https://doi.org/10.5066/P960N1F9 (2019).
This publication includes both .csv data files as well as metadata. A short description of these files and the
data found within them can be found in Tables 5 and 6. In addition, many additional soil attributes not
included in that publication, such as von Post decomposition index and additional soil chemistry
information, can be found for the majority of these data through various USGS Open-File Reports
(Manies et al., 2017; Manies et al., 2016; Manies et al., 2014; O'Donnell et al., 2013; O'Donnell et al.,
2012; Manies et al., 2004).

**6 Conclusions**
Boreal ecosystems are especially sensitive and vulnerable due to climate change. Models may not
accurately forecast high latitude biogeochemical processes for many reasons  (Flato et al., 2013). One
reason for the discrepancies between model results and data is that many models lack the input data
required, including important factors for modeling soil thermal dynamics like bulk density (Koven et al.,
2013; Khvorostyanov et al., 2008). While these processes are starting to be incorporated into land surface
and regional models (see, for example, Genet et al., 2013; Koven et al., 2011), currently few models
include the distinct properties of organic soils that are found in the boreal region (Flato et al., 2013). The
>3,000 soil samples, from >290 soil profiles, presented in this paper provide information regarding the
important soil properties of bulk density, C concentration, N concentration, C:N ratios, and thickness by
organic soil horizon. Such data are needed for initializing and validating models related to boreal organic
soils. In addition, these data can be used by scientists to calculate C and N stocks where researchers only
have soil horizon thickness data or to address shortcomings of missing data in instances when an
important soil property was not measured.


**Acknowledgements**

We would like to acknowledge the Bonanza Creek LTER and the USGS Fairbanks office for their support

of our work over the years. We would also like to thank the many people who assisted in collecting these

samples. This work, over the years, has been supported by the USGS (Land Resources, Climate and Land

Use Change, and Global Change programs), the National Science Foundation (DEB-0425328, EAR-

0630249), and the NASA Terrestrial Ecology (NNX09AQ36G).

**Author contribution**

KM prepared the manuscript with the help of MW and JH. All authors were involved in supporting the

collection of these data.

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

**Figure 1.** Location of the sites used in this study, all located within Interior Alaska. Regions, as ascribed in the dataset, are noted in red. Cities are written in yellow. (Map data: Google, 2020.)

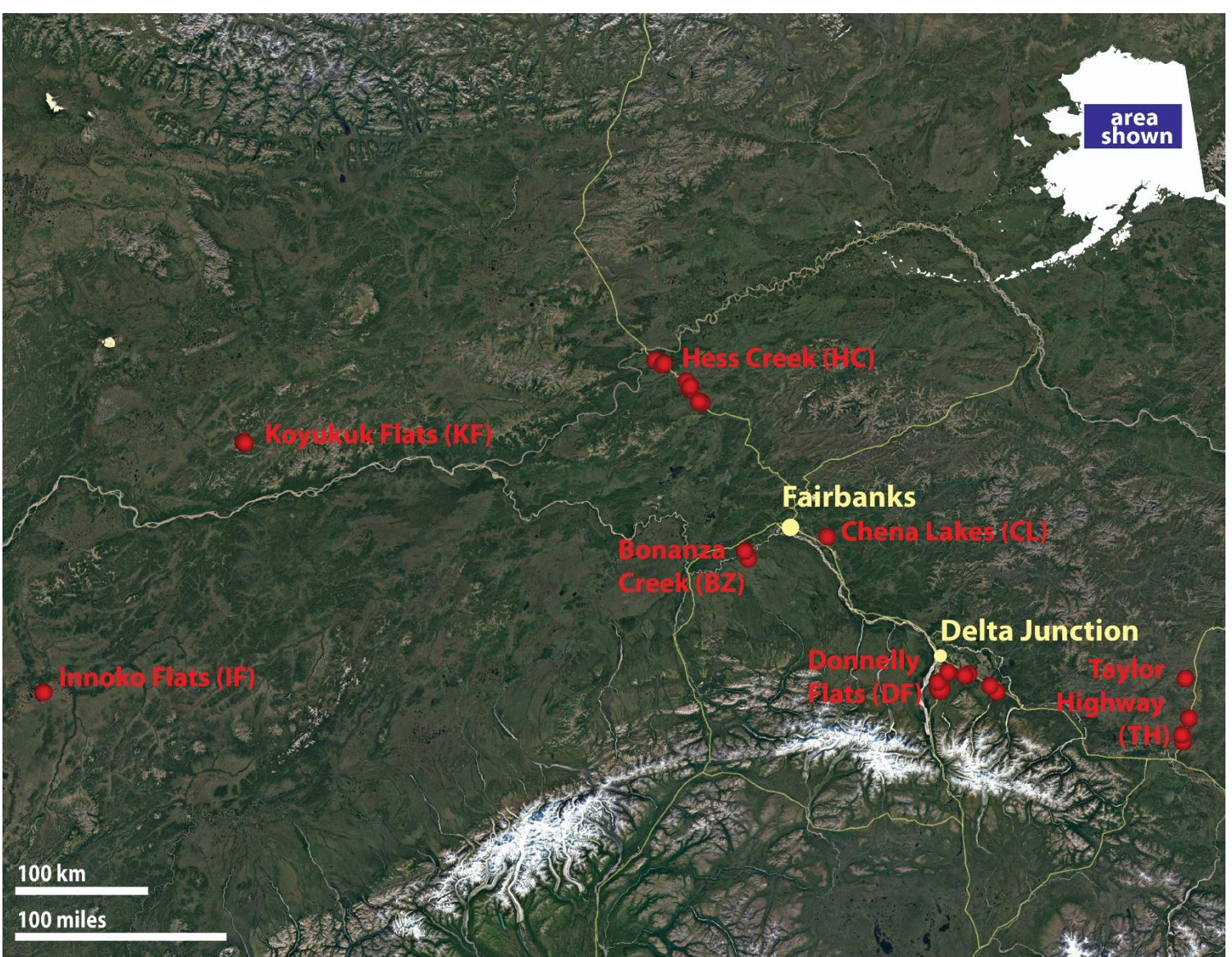

**Figure 2.** Soil drainage class decision tree. Beginning in the top left, if the soil meets the criteria, one has found the designated drainage class, having the characteristics located on the right. If the soil of interest does not meet the criteria, one moves down to the next drainage class to determine if its criteria is met. Drainage classes are also modified by slopes of greater than 5 % by moving up one drainage class.

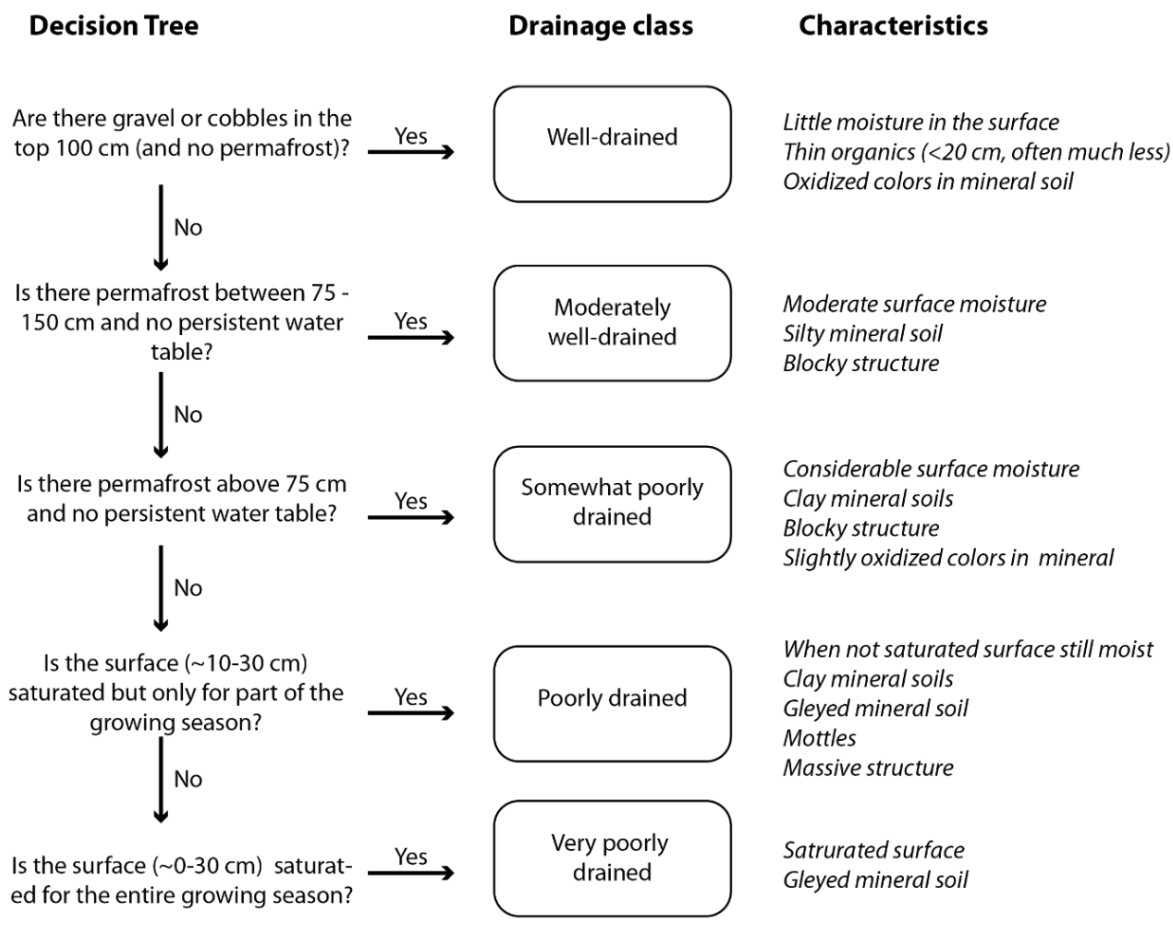

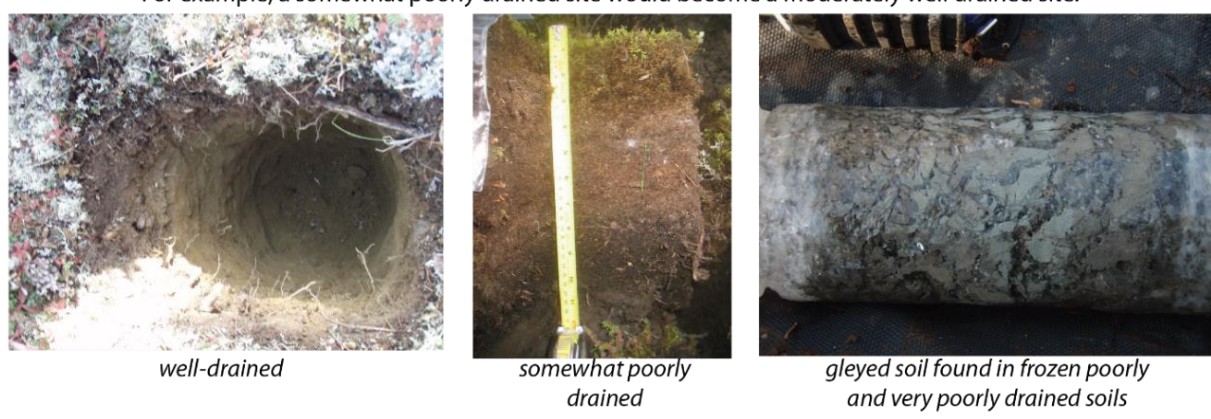

*well-drained*    *somewhat poorly drained*    *gleyed soil found in frozen poorly and very poorly drained soils*

**Figure 3.** Trends in carbon and nitrogen density (g cm$^{-3}$) by horizon type using average values for bulk density, carbon, and nitrogen (Table 2). Horizon designations: L = live moss, D = dead moss, F = fibric, M = mesic, H = humic, Min = mineral.

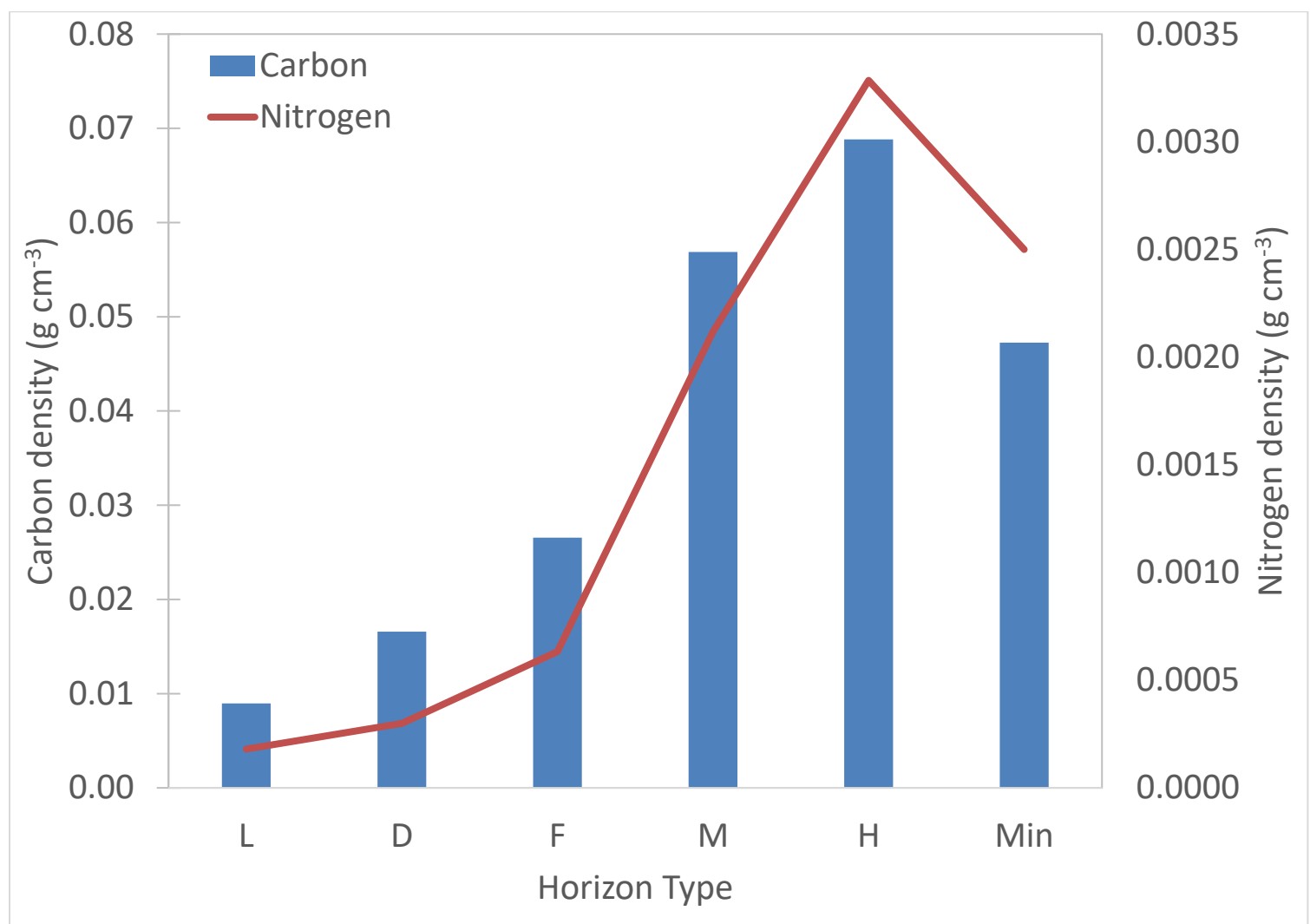

**Table 1.** A description of the soil horizons, as assigned by examining the composition of the soil horizon, including the degree of decomposition, color, and root abundance.

| Horizon Type | Horizon Code | Description |
|---|---|---|
| Live moss | L | Live moss, which is usually green. This horizon generally also contains a small amount of plant litter. Plant material is completely undecomposed. |
| Dead moss | D | Moss that is dead and either undecomposed or slightly decomposed. Plant parts are easily identifiable. This horizon would be considered an $O_i$ horizon in the U.S. soil system. |
| Fibric | F | Fibrous plant material that varies in the degree of decomposition (somewhat intact to very small plant pieces), but there is no amorphous organic material present. Very fine roots often make up a large fraction of this horizon. This horizon would be considered an $O_i$ horizon in the U.S. soil system. |
| Mesic | M | This horizon is comprised of moderately decomposed material, with few, if any, recognizable plant parts other than roots. There is amorphous present within this horizon to varying degrees, but it is not smeary. This horizon is often considered an $O_e$ horizon (U.S. soil system). |
| Humic | H | This organic horizon is highly decomposed and is mostly amorphous material. The soil in this horizon smears when rubbed and contains little to no recognizable plant parts. The H horizon is generally considered an $O_a$ horizon (U.S. soil system). |
| Mineral | Min | Classified as an A, B, or C mineral soil (U.S. soil system), it contains less than 20-volume-percent organic matter, as judged in the field. |

**Table 2.** Bulk density (g/cm$^3$), C (%), N (%), C:N ratio, and thickness (cm) for the main horizon codes averaged across all drainage and age classes. Significant differences (p < 0.05) among the main six horizon codes are indicated with different letters. There are no thickness values for mineral soil because these results would reflect the thickness sampled, not the actual thickness of this horizon. Stdev is one standard deviation.

| Horizon Code | Bulk Density (g/cm$^3$) | | | Carbon (%) | | | Nitrogen (%) | | | C:N | | | Thickness (cm) | | |
|---|---|---|---|---|---|---|---|---|---|---|---|---|---|---|---|
| | mean | stdev | n | mean | stdev | n | mean | stdev | n | mean | stdev | n | mean | stdev | n |
| live moss (L) | 0.022[a] | (0.018) | 138 | 41.7[a] | (3.8) | 145 | 0.84[a] | (0.25) | 145 | 53.8[a] | (16) | 141 | 2.5[a] | (1.6) | 136 |
| dead moss (D) | 0.039[b] | (0.026) | 540 | 42.6[a] | (3.8) | 538 | 0.77[a] | (0.27) | 537 | 62.1[a] | (23) | 541 | 13.9[b] | (24.2) | 157 |
| fibric (F) | 0.065[c] | (0.041) | 552 | 41.0[a] | (5.6) | 566 | 0.98[a] | (0.42) | 564 | 47.6[a] | (17) | 552 | 12.8[bc] | (17.9) | 221 |
| mesic (M) | 0.149[d] | (0.077) | 634 | 38.2[b] | (6.8) | 650 | 1.42[b] | (0.54) | 651 | 30.6[b] | (13) | 634 | 20.4[c] | (40.3) | 208 |
| humic (H) | 0.215[e] | (0.096) | 160 | 32.1[c] | (6.6) | 164 | 1.53[c] | (0.44) | 164 | 22.2[c] | (6) | 160 | 9.7[b] | (11.3) | 74 |
| mineral (Min) | 0.731[f] | (0.380) | 584 | 6.5[d] | (6.2) | 674 | 0.34[d] | (0.32) | 673 | 18.0[d] | (7) | 603 | -- | -- | -- |
| fibrous (D & F) | 0.052 | (0.037) | 1092 | 41.8 | (4.8) | 1104 | 0.88 | (0.37) | 1101 | 54.6 | (21) | 1101 | 22.8 | (41.1) | 220 |
| amorphous (M & H) | 0.162 | (0.085) | 794 | 36.9 | (7.2) | 814 | 1.44 | (0.52) | 815 | 28.9 | (12) | 813 | 19.7 | (27.7) | 263 |

**Table 3.** Thickness (cm) of the main horizon codes by soil drainage and age class. The mineral soil horizon was not included in this table because the way in which we sampled mineral soil led to arbitrary thicknesses. Significant differences (p < 0.05) for horizon codes among drainage classes are indicated with different letters. Stdev is one standard deviation.

| Horizon | | Drainage | | | | | Age | | |
| | | Well drained | Moderately Well Drained | Somewhat Poorly Drained | Poorly Drained | Very Poorly Drained | New | Young | Mature |
|---|---|---|---|---|---|---|---|---|---|
| live moss (L) | mean | 2.2$^a$ | 2.2$^a$ | 2.1$^a$ | 1.5$^a$ | 4.3$^b$ | 1.0$^{ab}$ | 2.6$^a$ | 2.4$^b$ |
| | stdev | (1.0) | (0.8) | (1.1) | (0.7) | (2.1) | (-) | (2.) | (1.2) |
| | n | 6 | 11 | 75 | 18 | 26 | 2 | 42 | 92 |
| dead moss (D) | mean | 3.3$^a$ | 7.4$^a$ | 7.6$^a$ | 6.5$^a$ | 38.1$^b$ | 6.3$^{ab}$ | 16.3$^a$ | 14.1$^b$ |
| | stdev | (1.6) | (6.8) | (10.8) | (6.5) | (40.8) | (4.5) | (20.0) | (27.5) |
| | n | 6 | 19 | 77 | 21 | 34 | 17 | 42 | 98 |
| fibric (F) | mean | 3.1$^a$ | 9.6$^{bc}$ | 7.9$^b$ | 13.7$^c$ | 40.2$^d$ | 6.4$^a$ | 19.8$^b$ | 14.0$^c$ |
| | stdev | (3.0) | (5.0) | (5.2) | (10.9) | (38.7) | (5.7) | (31.8) | (14.7) |
| | n | 11 | 18 | 121 | 45 | 26 | 65 | 39 | 117 |
| mesic (M) | mean | 2.8$^a$ | 13.3$^{ab}$ | 13.2$^{ab}$ | 14.4$^b$ | 57.2$^c$ | 6.3$^{ab}$ | 21.3$^a$ | 27.2$^b$ |
| | stdev | (1.3) | (17.6) | (38.0) | (21.8) | (53.4) | (3.5) | (33.2) | (50.8) |
| | n | 5 | 15 | 112 | 39 | 33 | 53 | 50 | 101 |
| humic (H) | mean | none | 12.1$^{ab}$ | 5.6$^a$ | 7.4$^a$ | 20.2$^b$ | 4.3$^a$ | 13.1$^{ab}$ | 11.9$^b$ |
| | stdev | -- | (15.4) | (7.4) | (3.4) | (14.7) | (3.2) | (12.7) | (13.1) |
| | n | -- | 7 | 38 | 13 | 16 | 24 | 17 | 33 |
| fibrous (D & F) | mean | 4.5$^a$ | 14.8$^b$ | 11.3$^b$ | 14.8$^b$ | 58.5$^c$ | 7.1$^a$ | 27.0$^a$ | 22.9$^b$ |
| | stdev | (4.4) | (8.1) | (9.9) | (11.2) | (47.8) | (6.3) | (36.7) | (26.9) |
| | n | 12 | 21 | 135 | 51 | 40 | 73 | 54 | 132 |
| amorphous (M & H) | mean | 2.8$^a$ | 15.8$^a$ | 14.3$^a$ | 16.1$^a$ | 63.2$^b$ | 7.7$^a$ | 23.0$^a$ | 29.9$^b$ |
| | stdev | (1.3) | (25.3) | (38.4) | (21.0) | (51.5) | (4.5) | (33.5) | (51.9) |
| | n | 5 | 18 | 118 | 41 | 35 | 56 | 56 | 105 |

**Table 4.** Physical and chemical properties of additional surface horizons. Number of observations, bulk density (g/cm$^3$), C (%), N (%), C:N ratio, and thickness (cm) of non-main horizon codes. Values in parenthesis are standard deviations.

| Horizon | N | Bulk density (g/cm$^3$) | Carbon (%) | Nitrogen (%) | C:N Ratio | Thickness (cm) |
|---|---|---|---|---|---|---|
| ash | 14 | 0.183 (0.155) | 38.0 (14.4) | 0.84 (0.34) | 49 (20) | 0.1 (-) |
| burned organics | 99 | 0.122 (0.142) | 38.6 (8.9) | 1.07 (0.32) | 99 (38) | 1.8 (1.0) |
| lichen | 31 | 0.034 (0.019) | 40.3 (5.9) | 0.76 (0.41) | 69 (37) | 4.1 (3.1) |
| litter | 16 | 0.044 (0.018) | 41.2 (3.1) | 1.55 (0.52) | 29 (10) | 1.6 (0.9) |

**Table 5.** Data columns found in megaAlaska_v11-2 for ScienceBase.csv. This datafile can be found at https://doi.org/10.5066/P960N1F9: Data Supporting Generalized models to estimate carbon and nitrogen stocks of organic layers in Interior Alaska.

| Column Name | Units | Column Description |
|---|---|---|
| sampleID | -- | The first four characters are based on the region and site. Then there is a space. Next the soil core number, followed by a period, and then the basal depth of the soil horizon. |
| depth | cm | Basal depth of the soil horizon |
| Hcode | -- | Horizon code as determined from Table 1 |
| Sample | -- | Qualitative description of the soil horizon |
| date | mm/dd/yy | Date sample was taken |
| thickness | cm | Thickness of the soil horizon |
| BDall | g/cm$^3$ | Bulk density, all soil |
| BDfine | g/cm$^3$ | Bulk density, fines (soil particles > 2 mm and roots > 1 cm diameter excluded) |
| HtAboveMin | cm | Height of each basal depth above the organic-mineral soil boundary |
| carbon | % | Carbon concentration |
| nitrogen | % | Nitrogen concentration |
| 13C | ‰ | Per mil (‰) value of delta $^{13}$C |
| 14C | ‰ | Per mil (‰) value of delta $^{14}$C for bulk soil sample |
| LOI | % | Loss-on-ignition value |
| volume_method | -- | Method used to sample soils volumetrically |
| region | -- | Region within Alaska where the site is located (Figure 1) |
| site | -- | Site where the core was taken |
| profile | -- | Soil profile, or core, number |
| drainage | -- | Soil drainage category (Figure 2) |
| standage | yrs | Age from last disturbance (fire or thaw) |
| ageclass | -- | N = newly burned (< 5 yrs), Y= young (5-50 yrs), M = mature (>50 yrs) |
| SurfaceVeg | -- | Types of vegetation found on the soil surface |
| SubbedBD | -- | If Y the bulk density is not a measured value. Instead an average value was used. |
| SubbedC | -- | If Y the carbon concentration is not a measured value. Instead an average value was used. |
| SubbedN | -- | If Y nitrogen concentration is not a measured value. Instead an average value was used. |
| GroupedHcode | -- | Horizon codes grouped into fewer categories |
| GroupedVeg | -- | Surface vegetation grouped into fewer categories |

**Table 6:** Data columns found in Site_GPS_coordinates_v2. This datafile can be found at https://doi.org/10.5066/P960N1F9: Data Supporting Generalized models to estimate carbon and nitrogen stocks of organic layers in Interior Alaska.

| Column Name | Description |
| --- | --- |
| Region | Region within Alaska where the site is located (Figure 1) |
| Region Code | Two letter code for the region |
| Site | Site where the core was taken |
| Profile | Which soil profiles are located at this location - all indicates general coordinates for all soil profiles |
| Latitude | Latitude in decimal degrees |
| Longitude | Longitude in decimal degrees |
| Datum | Datum of the coordinates |