# Peer review of "Generalized models to estimate carbon and nitrogen stocks of organic soil horizons in Interior Alaska"

_Earth System Science Data, 2019_

## Referee Comment (RC1) · Anonymous Referee #1 · 18 Feb 2020

Manies et al. present a soil carbon and nitrogen data set for boreal Alaska and categorize the soil profiles according to their drainage conditions and disturbance history. The methods of collection and analysis of the data are well explained and this manuscript presents a very valuable data set which is worth publishing. I suggest to publish this manuscript after minor revisions. In particular, it would be beneficial to have a more extended discussion of the data set and how this data set relates to other soil carbon and nitrogen data sets in Alaska. Please find below more detailed comments on the manuscript.

Abstract

Line 3: I suggest reformulating the sentence “Boreal soils [. . .] organic soils”. E.g. “Boreal soils are unique because the mineral soil layers are often covered by thick layers of organic soil”

Line 4: Is there a difference between “layers” and “horizons”? You use both terms throughout the entire manuscript. I suggest you define that in the method parts or stick to one of these expressions.

Introduction

Detailed comments:

Line 16: Please reformulate the sentence “These soils [. . .] organic soil layer.” In particular, avoid writing “in that” since this makes the sentence structure complicated.

Line 22: Please better link the last sentence of this paragraph to the rest of the paragraph or highlight the importance of it. E.g. “. . .especially where protected by permafrost, which make boreal soils high C containing soils including mineral and organic layers”

Line 24: Please add a source for the first sentence of the paragraph

Line 24: Replace “is” with “are”

Line 30: layers or horizons?

Line 35: I suggest writing “C and N” instead of C/N. This might be misleading. C/N is often understood as C:N ratio.

Line 36: I suggest writing “Fires affect. . .”

Line 36: I suggest replacing “several” with “multiple”

Line 37: “First” but where is the second and third in this paragraph? – Please restructure the paragraph and make it more clear, which are the several ways boreal soils are affected by fire.

Line 37: I suggest reformulating “the amount of which”. E.g. The soil depth, which is affected, varies based on the fire severity

Line 38: Add “Second, the. . .”

Line 41: How does post-fire vegetation affect the chemistry of C and N inputs to the soil? Please add another sentence or give an example.

Line 47/48. This is a hard transition from these two paragraphs. Please try to better link these paragraphs. E.g. Boreal soils have not been adequately characterized in terms of their disturbance history.

Line 52: I suggest writing “including more than 3000 observations” instead of “(> 3000 observations)”

Line 53: I suggest writing “categorized” instead of “examined”

Line 54: I suggest writing “data set” instead of “results”

Methods

General comments:

I suggest adding a sentence over which time period the samples were collected. Could you add information on the depth of sampling? What was the aimed depth during sampling? How many locations and sites were sampled? In the data set, I find 57 different locations with coordinates but in chapter 3.7 it is written that more than 290 soil profiles were sampled. Please state that in the text.

I think it is great that you established a classification for soil drainage.

Detailed comments:

Line 77: I suggest naming this classification as Table 1.

Line 93/94: The references here are basically the same. Soil Survey Staff (1982, 1993), just different editions. I suggest choosing one and naming it Soil Survey Staff

(1993) in the reference.

Line 98: I suggest replacing “in that” with “because”

Line 101: Redoximorphic – check the spelling

Line 136: “In the field the best call was made to if it was. . .” I suggest re-writing this sentence. E.g. “in the field visual inspection of the soil samples gave a first indication. . .”

Line 140: Please add “R core Team” to the source. Also, I could not find this reference in the reference list

Line 142: “was tested” Please add “for significant differences among the different soil horizons”

Dataset Review

General comments:

In this section you talk about horizon or horizon types. Is this different to layers?

I suggest adding another main chapter for the two subchapters 3.6 and 3.7 since it is more a discussion chapter. I suggest adding “4. Discussion of the data set” and then include the chapters 3.6 and 3.7 in there. In addition, my question is, how does your data set relate to the Soil pedon carbon and nitrogen data for Alaska by Michaelson et al. (2013). Maybe you can refer to that during the discussion and indicate how your data set adds or fits within this data set. Also, in a discussion chapter you could state again why your data set is so valuable.

Detailed comments:

Line 169: Please write “were not” instead of “weren′t”

Line 178: “likely due in large. . .” I suggest rephrasing this sentence.

Line 204: I suggest naming this sub-chapter “soil horizon thickness”

Line 215: Please write “was not” instead of “wasn't”

Line 216: Please write “it is” instead of “it's”

Line 238: I wonder whether the accuracy of the bulk density measurements could be a reason for the differences, too. Often it is difficult to accurately measure bulk density, therefore I could think that maybe the accuracy of the bulk density measurements in both, the reference and your data set might be a reason for the differences.

Line 239: I do not understand the first sentence. What do you mean with “previous results”? Please, also consider restructuring this sentence.

Line 246: Please insert a “the” between “than” and “measured”

Line 261: Please insert a “the” between “into” and “mineral”

Line 264: Please add Boreal in this sentence.

Line 264: I suggest moving the first sentence of this paragraph to the conclusions.

Line 265: I suggest writing “drained sites” instead of “drainages”.

Line 274: Please change “is” to “are”

Conclusions

Line 282: I suggest writing “lack input data for” instead of “do not do a good job”

Figures and Tables

Figure 1: I suggest writing “included” instead of “used” in the figure caption. I suggest to improve the map because the scale bar is hardly visible and the map looks a bit blurred in general. Maybe add more names in the maps for the regions or locations where the sites are.

Figure 2: I like this figure. However I suggest to add photographs with a higher resolution.

Figure 3: This figure shows that humic soils are most important in C and N storage. Maybe you can mention this in the text as well.

Table 1 + 2: I have some troubles understanding the tables and whether it is significant different or not. There are a lot of superscript letters, sometimes the same, sometimes two or three. While I acknowledge the effort in putting everything into one table, I would suggest to make a separate cross-table for the p values and whether it is significant different or not. The same for the tables S2-S5 in the supplementary material.

Dataset:

Thank you for publishing this very valuable data set.

Generalized_models_for_C&N_Alaska.csv: You write in the methods section that four different methods were used in collecting the soil cores. I suggest adding a column to the data set indicating which method was used for the collection of the samples.

Site_GPS_coordinates.csv: Could you add the key for the abbreviations of the regions and sites. I could not find it in the metadata what e.g. HCCS or BZ means. Also, it would be nice if the sites could be found in Figure 1, e.g. by adding the region names to the map.

References:

Michaelson, G.J., Ping, C.-L., and Clark, M. (2013). Soil Pedon Carbon and Nitrogen Data for Alaska: An Analysis and Update. Open Journal of Soil Science 3, 132-142. doi: 10.4236/ojss.2013.32015

---

## Referee Comment (RC2) · Anonymous Referee #2 · 6 Apr 2020

The authors describe an 'aggregated' soil carbon and nitrogen dataset for boreal Alaska. Aggregation of the field measured data, for over soil 289 profiles and over 3000 samples (not provided here), is according to soil drainage, state of decomposition and time since last disturbance. This is a valuable contribution since, overall, organic horizons of boreal soils have seldom been adequately characterized (i.e. most studies focused on the mineral part). Soil laboratory methods are well documented, and the chemical methods and statistical methods well described; possible limitations of the data are appropriately discussed. For example, the study illustrates some of the issues that may arise when disparate data sets (i.e. different methods/standards) are used (e.g. line 248-255).

[Figure]

The description of the dataset itself in the manuscript is rather limited and this should be improved by adding an Appendix (see comments under data sets below).

Please find below some other comments:

Manuscript: line 19: Add some more town names to Figure 1. Add footnote that several profiles were sampled at a given site. Include more visible North arrow and scale bar (in SI units rather than miles).

line 20: Picture bottom right. Legend includes two drainage classes? How have such cases, when arising, been processed in the dataset itself?

line 35: change C/N to C and N dynamics to avoid possible confusion with C:N ratio.

line 37: There is no second or third, please rephrase this paragraph.

line 52: over 3000 thousand

line 77: Possibly present this as a table (new Table 1): code (X), name, description.

line 156: one progresses

line 157: express bulk density as g cm-3.

line 187 and 188: Both horizon and layer are used. Are they used as synonyms or were (thicker) horizons divided into separate layers for sampling? Please clarify this.

line: 190: than then → (change to) than the

line 202: Change Thickness to Horizon thickness.

line 222: I would suggest you move subsection 3.6 and 3.7 to a new Discussion section (4). This section could also include a comparison with results derived from other studies for boreal regions.

line 274-278: The authors should at least indicate that the units of measurement, respectively domains for observations, for the properties under consideration in the two

csv-files can be found in file Mega-AK metadata.xml. However, I would recommend this information is also summarised in an Appendix.

As indicated, it would be a 'plus' for this data paper if the underpinning 'raw' profiles could also be made available as supplemental information, as they are not presented in https://doi.org/10.5066/P960N1F9, rather than just pointing at several open file reports (pdf's) as is now the case. General: Abbreviations like weren't, wasn't etc do not belong in a manuscript. Please recheck the whole document for such instances.

Figure 3: What depth interval is considered in this figure? The unit of g/cm2 is rather confusing (a typo?).

Table 1: Bulk density given as g/cm2, this should be g/cm3.

Table 1 - 4: For legibility, and future typesetting by ESSD, it would be better to create three columns for each row (e.g. bulk density): n, mean, sd. The symbols for statistical significance would then also become more 'legible'.

Table 3: Bulk density, should be g/cm3. Is this a systematic typo or something else?

Datasets:

Under 'Data access' briefly describe the content of the zip file (csv and xml). Further, please provide an Appendix that describes the content of the csv files.

- Site_GPS_coordinates.csv:

There are 57 sites, yet the paper refers to over 289 profiles. It would be useful to know how many profiles were sampled at each site without readers having to digest this from file (Generalized_models_for_C&N_Alaska.csv). Further, the abbreviations for regions and sites should be provided, preferably in a look-up table (i.e. as a separate csv file). Please note that data in row 57 have 'shifted' to the right; this should be corrected.

- Generalized_models_for_C&N_Alaska.cvs

Specify units of measurement (depth (cm), bulk density (g/cm3), 13C etc.); explain all codes/abbreviations used in the file, as a 'look up' table (i.e. as a separate csv file). For example, add the descriptions from Fig. 2 for drainage.

Note: I realise this information is largely stored in file 'Mega-AK metadata.xml', but this is not really user friendly for the reader. Further, for some properties the measurement units are not specified (see e.g. LOI).

Supplemental information S2: Please add units for bd table.

---

## Author Response (AR1)

**Response to Reviewer 1**

(line numbers refer to original manuscript/current manuscript)

line 3/3: I suggest reformulating the sentence "Boreal soils […] organic soils. *I tried rewriting this sentence several times and have a hard time rewording it as suggested so I am leaving the text as it was.*

Line 4/36: Is there a difference between "layers" and "horizons"? You use both terms throughout the entire manuscript. I suggest you define that in the methods or stick to one of these expressions. *Layers and horizons are the same thing. This is now clarified in line 36. In addition, the text of the manuscript was changed so that horizon is consistently used throughout.*

Line 16/16: Please reformulate the sentence "These soils [….] organic soil layer. *Sentence simplified.*

Line 22/22: Please better link the last sentence of this paragraph to the rest of the paragraph or highlighting the importance of it. *A sentence has been added to link the last sentence to the rest of the paragraph: "Thus, both organic and mineral soil play an important role determining the amount of C stored in boreal ecosystems."*

Line 24/24: Please add a source for the first sentence of the paragraph. *Reference added. In addition, the text of this entire paragraph has been strengthened and additional references have been added.*

Line 24/n.a.: Replace "is" with "are": *This text no longer exists (see previous point).*

Line 30/36: layers or horizons? *This issue is now corrected (see line 4 response).*

Line 35/41: I suggest writing "C and N" instead of C/N. This might be misleading. C/N is often understood as C:N ratio. *This entire sentence has been rewritten so this request no long applies (see next response).*

Line 36-48/41-62: I suggest writings "Fires affect…"; I suggest replacing "several" with "multiple"; "First" but where is the second and third in this paragraph. Please restructure the paragraph and make it more clear, which are the several ways boreal soils are affect by fire; I suggest reformulating "the amount of which". *This paragraph is now rewritten with these comments incorporated and now reads:*

> The main disturbances that affect boreal soil properties are fire and permafrost thaw. Fires affect boreal soils through the combustion of litter and surface organic layers (as ground fuel; Harden et al., 2000), with the amount and depth of combustion regulated by fire severity (Turetsky et al., 2011). Fire directly effects surface organic soils, both in elemental composition and structure (Neff et al., 2005). In addition, there are indirect effects of fire on soil properties. The loss of insulating organic soil results in a darkened soil surface, which in turn warms post-fire soils, increasing decomposition rates from the surface downward (Genet et al., 2013; O'Neill et al., 2002). In addition, both fire return interval and fire severity influence post-fire vegetation and the re-accumulation of organic soil layers. As different tree and understory species have different amounts of C and N in their tissues (Van Cleve et al., 1983), changes in post-fire vegetation affect soil C and N accumulation rates and thus, the concentration of these elements in surface soil. Permafrost thaw also affects soil properties in several ways. By definition, thaw exposes older, previously sequestered C to warmer soil temperatures (Osterkamp et al., 2009), increasing rates of decomposition (Mu et al., 2016; Schadel et al., 2016). In well drained sites post-thaw conditions usually result in water draining from the soil, resulting in oxic conditions (Estop-Aragonés et al., 2018). In lowlands, permafrost thaw often results in subsidence and inundation, changing the ecosystem from a forested permafrost plateau to a thermokarst wetland (Schuur et al., 2015). Fire can often be a trigger for this rapid permafrost thaw (Myers-Smith et al., 2008). Post fire vegetation changes affects both C and N inputs, again affecting the concentration of these elements within surface organic soil layers. As both fire frequency and permafrost thaw are expected to increase in the future (Hinzman et al., 2005), biogeochemical models have a need to characterize how these disturbances will impact C and N stocks. To accurately represent future scenarios, models need to include the distinct properties of organic soil horizons found in the boreal region (Flato et al., 2013).

Line 41/49: How does post-fire vegetation affect the chemistry of C and N inputs to the soil? Please add another sentence of give an example. *This paragraph is now rewritten, with this issue addressed. Please see above response.*

Line 47-48/58-60: This is a hard transition from these two paragraphs. Please try to better link these paragraphs: *This paragraph is now rewritten, with this issue addressed. Please see above response.*

Lines 52/71: I suggest writing "more than 3000 observations" instead of "> 3000 observations". *Wording changed.*

Line 62/79-87: I suggest adding a sentence over which time period the samples were collected. Could you add information on the depth of sampling? How many locations and sites were sampled? In the data set I find 57 different locations with coordinates but in chapter 3.7 it is written that more than 290 soil profiles were sampled. Please state that in the text. *The beginning of the field methodology section now gives details on the number of sites, cores, and profiles taken as well as discusses the depth of sampling. This paragraph now reads*

Soil cores were sampled at 58 different sites located within several areas of Interior Alaska (Figure 1). Several different ecosystem types were sampled, including black spruce forests (~50%); wetlands (~26%); and deciduous and mixed forests (~16%). Between 1 and 14 soil profiles were sampled at each site, for a total of 292 soil profiles. Sampling took place over a 15-year period from 2000-2015. We examined the effect of fire or permafrost thaw disturbance on soil properties by categorizing each of the soil profiles in relation to time since the last disturbance, which we divided into three age classes: new (<5 yrs old), young (5 – 50 yrs old), and mature (> 50 yrs old). All new sites were recently burned and thus had lost some portion of their surface organic horizons (Harden et al., 2000), while young sites experienced either fire or permafrost thaw.

Line 77/130: I suggest naming this classification as Table 1. *This information is now included as a table. In addition, we added more text to Table 1 to expand upon these definitions. The manuscript gives very brief descriptions of the six main horizons: "A description of the horizons and the codes we used to represent them are found in Table 1, but in summary there are six main horizons: live moss (L), dead moss (D), fibric (mostly undecomposed; F), mesic (more decomposed; M), humic (very decomposed; H), and mineral soil (Min)."*

Line 93-94/132-134: The references here are basically the same, just different editions. I suggest choosing one and naming it Soil Survey Staff (1993) in the reference. *One of these references is for the US systems, while the other is for a Canadian manual so both references are still included. The references have been fixed in the citation program to provide the entire author name, making this fact clear.*

Line 98/94: I suggest replacing "in that" with "because". *We prefer the original wording.*

Line 101/98: Redoximorphic – check the spelling. *Spelling corrected*

Line 136/175: "In the field the best call was made to if it was…" I suggest re-writing this sentence something like "in the field visual inspection of the soil samples gave a first indication." *Rewritten as "Using visual and textural cues the field, horizons were categorized as either mineral (< 20 % C) or organic (≥ 20 % C)."*

Line 140/180: Please add "R core Team" to the source. Also, I could not find this reference in the reference list. *This reference was included, but the way in which it was entered into the citation program had it appearing as 'Team, RC'. This citation is now fixed.*

Line 142/183: Please add "for significant differences among the different soil horizons" after "was tested". *Verbiage added.*

Line 169/208: Please write "were not" instead of "weren't". *Sentence rewritten changed.*

Line 178/223: "likely due in large…" I suggest rephrasing this sentence. *Wording changed.*

Line 204/250: I suggest naming this sub-chapter "soil horizon thickness". *Heading title changed.*

Line 215/262: Please write "was not" instead of "wasn't". *Wording changed.*

Line 216/263: Please write "it is" instead of "it's". *Wording changed.*

Line 222/270: I suggest adding another main chapter for the two subchapters 3.7 and 3.7 since it is more a discussion chapter. I suggest adding 4. Discussion of the data set" and then include 3.6 and 3.7 there. In addition, my question is, how does your data set relate to the soil pedon carbon and nitrogen data for Alaska by Michaelson et al. (2013). Maybe you can refer to that during the discussion and indicate how your data set adds or fits within this data set. Also, in a discussion chapter you could state again why your data set is so value. *The headings were changed as suggested. In addition, you ask how our data compares to Michaelson et al, 2013. This paper, as well as Ping et al, 2010, are now brought up in the Introduction (lines 66-69) as well as we now discuss how our data compare to Michaelson (2013) and Ping (2010) in section 4.1:*

> Our data are the first of its kind to present organic horizon data across a range of Alaskan boreal ecosystems.  Other studies have examined organic soil as a separate entity from mineral soil but with certain limitations. Michaelson et al. (2013) used Alaskan USDA-NRCS soil pedon data to examine soil properties of both organic and mineral soil but present these data for the organic portion as a whole. This study shows that there is significant variation in bulk density and C and N concentration across organic horizons, and therefore, one should not disregard these horizon-based variations.  In a separate study, Ping et al. (2010) separated the organics into two horizons from boreal black spruce stands (Osurface, Oe/Oa). Our study supports the results of Ping et al., (2010), which found a decrease in C:N ratios with increasing depth. Moreover, our study provides data from a fuller suite of soil horizons and includes data from bogs, fens, and deciduous forests.

Line 238/297 & 312: I wonder whether the accuracy of the bulk density measurements could be a reason for the differences, too. Often it is difficult to accurately measure bulk density, therefore I could think that maybe the accuracy of the bulk density measurements in both, the reference and your data set might be a reason for the differences. *Yes, bulk density measurements are hard to get. This idea is now acknowleged in both paragraphs of section 4.2: "In addition, accurate bulk density measurements is time consuming to do correctly (Nalder and Wein, 1998) and could also play a role." And "Thus, bulk density measurements play a role in these differences."*

Line 239/299: I do not understand the first sentence. What do you mean with "previous results"? Please, also consider restructuring this sentence. *This wording is now changed to be clearer and reads "To determine if the above findings…".*

Lines 246, 261, 264, 265, 274, and 282: Please insert a "the" between "than" and "measured"; Please insert a "the" between "into" and "mineral"; Please add Boreal in this sentence; I suggest writing "drained sites" instead of "drainages"; Please change "is" to "are"; I suggest writing "lack input data for" instead of "do not do a good job". *Wording changed.*

Line 264/330: I suggest moving the first sentence of this paragraph to the conclusions. *This paragraph has been rewritten.*

Figure 1: I suggest writing "included" instead of "used" in the figure caption. I suggest to improve the map because the scale bar is hardly visible and the map looks a bit blurred in general. Maybe add more names in the maps for the regions or locations where the sites are. *Region names and additional city added to the figure. Scale text enlarged and made white to improve readability. Resolution of image made the maximum size possible. Figure caption modified accordingly.*

Figure 2. I like this figure. However, I suggest to add photographs with a higher resolution. *The size and resolution of the photographs have been increased. In addition, if helpful, we can provide the original photographs to the journal.*

Figure 3. This figure shows that humic soils are most important in C and N storage. Maybe you can mention this in the text as well. *The role of lower organic horizons in C storage is now called out in section 3.2: "Therefore, even though the deeper organic horizons (M and H) have slightly lower C concentrations than the shallow horizons, their high bulk densities result in large amounts of C at depth. In fact, given average thickness, bulk density, and % C (Table 2), approximately 75% of the soil C is stored in the mesic and humic soil horizons."*

Table 1 & 2 (now Table 2 & 3) plus Table S2-S5: I have some troubles understanding the tables and whether it is significant different or not. There are a lot of superscript letters, sometimes the same, sometimes two or three. While I acknowledge the effort in putting everything into one table, I would suggest to make a separate cross-table for the p values and whether it is significant different or not. The same for the tables S2-S5 in the supplementary material. *The formatting of these tables has been redone to help make comparison of the superscript letters easier.*

Thank you for publishing this very valuable data set. *Thank you for your suggestions on how to make this manuscript better.*

Generalized_models_for_C&N_Alaska.csv: You write in the methods section that four different methods were used in collecting the soil cores. I suggest adding a column to the data set indicating which method was used for the collection of the samples. *This information is now a part of the dataset (see Table 5).*

Site_GPS_coordinates.csv: Could you add the key for the abbreviations of the regions and sites. I could not find it in the metadata what e.g. HCCS or BZ means. Also, it would be nice if the sites could be found in Figure 1, e.g. by adding the region names to the map. *This information is now added to the file (see Table 6) and mentioned in Figure 1.*

**Response to Reviewer #2**

Line 19/Figure 1: Add some more town names to Figure 1. Add footnote that several profiles were sampled at a given site. Include more visible North arrow and scale bar (in SI units rather than miles). Region names and additional city added to the figure. *This information is now included.*

Line 20/Figure 2: Picture bottom right. Legend includes two drainage classes? How have such cases, when arising, been processed in the dataset itself? *The caption for Figure 2 is now modified to make it clear that gleyed soil can be found in both very poorly and poorly drained soil. The determining factor between the two classes is the length of time of saturated soils, as mentioned in the table.*

Line 35/41: Change C/N to C and N dynamics to avoid possible confusion with C:N ratio. *Sentence rewritten*

Line 37/41-62: There is no second or third, please rephrase this paragraph. *Paragraph rephrased. Please see the revised text in response to Reviewer 1 above.*

Line 52/71: over 3000 thousand. *Text changed.*

Line 77/Table 1: Possibly present this as a table (new Table 1): code (X), name, description. *This information is now presented as Table 1.*

Line 156/195: one progresses. *Sentence rewritten*

Line 157/196: express bulk density as g cm-3. *Text changed.*

Line 187/36: Both horizon and layer are used. Are they used as synonyms or were (thicker) horizons divided into separate layers for sampling? Please clarify this. *Layers and horizons are the same thing. This is now clarified in line 36, but the text of the manuscript was changed so that horizon is consistently used throughout.*

Line 190/235: than then → (change to) than the. *Text changed.*

Line 204250: Change Thickness to Horizon thickness. *This header has been changed to "Soil horizon thickness".*

Line 222/270: I would suggest you move subsection 3.6 and 3.7 to a new Discussion section (4). This section could also include a comparison with results derived from other studies for boreal regions. *The headings were moved as suggested. In addition, we have added a comparison of our study to the two other studies we know of that discuss organic soil properties in section 4.1. (See response to Reviewer #1, line 222 above.)*

Line 274 – 278/336-343: The authors should at least indicate that the units of measurement, respectively domains for observations, for the properties under consideration in the two csv-files can be found in file Mega-AK metadata.xml. However, I would recommend this information is also summarised in an Appendix. As indicated, it would be a 'plus' for this data paper if the underpinning 'raw' profiles could also be made available as supplemental information, as they are not presented inhttps://doi.org/10.5066/P960N1F9, rather than just pointing at several open file reports (pdf's) as is now the case. *There is now a short description of the data found within the ScienceBase publication ("This publication includes both .csv data files as well as metadata. A short description of these files and the data found within them can be found in Tables 5 and 6.") as well as two Tables that list the column names, units (if applicable), and column descriptions. I would also like to clear that all raw data used in this study are within this ScienceBase publication. The reason to point the reader to the Open-File Reports is because there is additional information (such as Von Post descriptions and $^{210}$Pb data) that may be of interest that are not in the ScienceBase publication.*

Figure 3: What depth interval is considered in this figure? The unit of g/cm2 is rather confusing (a typo?). *This graph is of C and N density and was incorrectly labeled. Therefore, depth or thickness is not included in the calculation. Thank you very much for spotting this labeling error. The figure caption and graph axes have been fixed.*

Table 1: Bulk density given as g/cm2, this should be g/cm3. *Units fixed.*

Tables 1-4: For legibility, and future typesetting by ESSD, it would be better to create three columns for each row (e.g. bulk density): n, mean, sd. The symbols for statistical significance would then also become more 'legible'. *The formatting has been redone for these tables to help make comparison of the superscript letters easier.*

Table 3 (now Table 4): Bulk density, should be g/cm3. *Units fixed.*

Under 'Data access' briefly describe the content of the zip file (csv and xml). Further, please provide an Appendix that describes the content of the csv files. *There are now two tables (5 & 6) describing that data available in the two .csv files.*

Site_GPS_coordinates.csv: There are 57 sites, yet the paper refers to over 289 profiles. It would be useful to know how many profiles were sampled at each site without readers having to digest this from file (Generalized_models_for_C&N_Alaska.csv). Further, the abbreviations for regions and sites should be provided, preferably in a look-up table (i.e. as a separate csv file). Please note that data in row 57 have 'shifted' to the right; this should be corrected. *We do not see a good place to put the number of cores per site in the text and feel that, if needed, this information is accessible using the data file. Region abbreviations in addition to names are now included in this file (see Table 6). Site names do not have much meaning outside of the research group, so we did not create a lookup table for this information.*

Generalized_models_for_C&N_Alaska.cvs. Specify units of measurement (depth (cm), bulk density (g/cm3), 13C etc.); explain all codes/abbreviations used in the file, as a 'look up' table (i.e. as a separate csv file). *This information is now included in Table 5. Tables 1, 5, and 6 have been added to the ScienceBase publication site as .txt files.*

Supplemental information S2: Please add units for bd table. *Units fixed.*

**Additional improvements**

Since our initial submittal of this manuscript we have come to realize that the original test used to determine differences among drainage types or age class, *difflsmeans*, does not correct for multiple comparisons. Therefore, we redid our analyses using estimated marginal means (*emmeans*; line 185), which does this correction, the result of which made some of significant differences originally presented in Tables 2-3 and Tables S2-S5 no longer significant.  This change did not alter our conclusions in any way.

To aid those interested in better understanding the predicted versus measured relationships discussed in section 4.2 we have added graphs showing those results to the Supplemental Information as Figures S1 and S2.

We have added a paragraph (line 318) suggesting that, due to the inherent variability of thickness measurements, that we recommend that researchers continue to measure thickness at their sites and only use our bulk density and concentration data. This combination minimizes errors while allowing researchers, if needed, to bypass soil sampling and processing, both of which are quite labor intensive, and, thus, not always possible. The new text is below:

[revised manuscript text omitted]

**Decision Tree**

**Drainage class**

**Characteristics**

Are there gravel or cobbles in the top 100 cm (and no permafrost)? —Yes→ **Well-drained**

*Little moisture in the surface*
*Thin organics (<20 cm, often much less)*
*Oxidized colors in mineral soil*

↓ No

Is there permafrost between 75 - 150 cm and no persistent water table? —Yes→ **Moderately well-drained**

*Moderate surface moisture*
*Silty mineral soil*
*Blocky structure*

↓ No

Is there permafrost above 75 cm and no persistent water table? —Yes→ **Somewhat poorly drained**

*Considerable surface moisture*
*Clay mineral soils*
*Blocky structure*
*Slightly oxidized colors in mineral*

↓ No

Is the surface (~10-30 cm) saturated but only for part of the growing season? —Yes→ **Poorly drained**

*When not saturated surface still moist*
*Clay mineral soils*
*Gleyed mineral soil*
*Mottles*
*Massive structure*

↓ No

Is the surface (~0-30 cm) saturated for the entire growing season? —Yes→ **Very poorly drained**

*Satrurated surface*
*Gleyed mineral soil*

**Slope modifier:**
If the slope of the site is greater than 5% the site should be better drained by one drainage class.
For example, a somewhat poorly drained site would become a moderately well drained site.

[Figure]

[Figure]

[Figure]

[Figure]

**Figure 2.** Soil drainage class decision tree. Beginning in the top left, if the soil meets the criteria, one has found the designated drainage class, having the characteristics located on the right. If the soil of interest does not meet the criteria, one moves down to the next drainage class to determine if its criteria is met. Drainage classes are also modified by slopes of greater than 5 % by moving up one drainage class.

[Figure]

**Decision Tree**

Are there gravel or cobbles in the top 100 cm (and no permafrost)? —Yes→

No ↓

Is there permafrost between 75 - 150 cm and no persistent water table? —Yes→

No ↓

Is there permafrost above 75 cm and no persistent water table? —Yes→

No ↓

Is the surface (~10-30 cm) saturated but only for part of the growing season? —Yes→

No ↓

Is the surface (~0-30 cm) saturated for the entire growing season? —Yes→

**Drainage class**

Well-drained

Moderately well-drained

Somewhat poorly drained

Poorly drained

Very poorly drained

**Characteristics**

Little moisture in the surface
Thin organics (<20 cm, often much less)
Oxidized colors in mineral soil

Moderate surface moisture
Silty mineral soil
Blocky structure

Considerable surface moisture
Clay mineral soils
Blocky structure
Slightly oxidized colors in mineral

When not saturated surface still moist
Clay mineral soils
Gleyed mineral soil
Mottles
Massive structure

Satrurated surface
Gleyed mineral soil

Slope modifier:
If the slope of the site is greater than 5% the site should be better drained by one drainage class.
For example, a somewhat poorly drained site would become a moderately well drained site.

[Figure]

well-drained          somewhat poorly          gleyed soil found in frozen poorly
                      drained                  and very poorly drained soils

**Figure 3.** Trends in carbon and nitrogen density (g cm⁻³) by horizon type using average values for bulk density, carbon, and nitrogen ( Table 2). Horizon designations: L = live moss, D = dead moss, F = fibric, M = mesic, H = humic, Min = mineral.

[Figure]

[Figure]

**Table 1.** A description of the soil horizons, as assigned by examining the composition of the soil horizon, including the degree of decomposition, color, and root abundance.

| Horizon Type | Horizon Code | Description |
|---|---|---|
| Live moss | L | Live moss, which is usually green. This horizon generally also contains a small amount of plant litter. Plant material is completely undecomposed. |
| Dead moss | D | Moss that is dead and either undecomposed or slightly decomposed. Plant parts are easily identifiable. This horizon would be considered an $O_i$ horizon in the U.S. soil system. |
| Fibric | F | Fibrous plant material that varies in the degree of decomposition (somewhat intact to very small plant pieces), but there is no amorphous organic material present. Very fine roots often make up a large fraction of this horizon. This horizon would be considered an $O_i$ horizon in the U.S. soil system. |
| Mesic | M | This horizon is comprised of moderately decomposed material, with few, if any, recognizable plant parts other than roots. There is amorphous present within this horizon to varying degrees, but it is not smeary. This horizon is often considered an $O_e$ horizon (U.S. soil system). |
| Humic | H | This organic horizon is highly decomposed and is mostly amorphous material. The soil in this horizon smears when rubbed and contains little to no recognizable plant parts. The H horizon is generally considered an $O_a$ horizon (U.S. soil system). |
| Mineral | Min | Classified as an A, B, or C mineral soil (U.S. soil system), it contains less than 20-volume-percent organic matter, as judged in the field. |

**Table 2.** Bulk density (g/cm³), C (%), N (%), C:N ratio, and thickness (cm) for the main horizon codes averaged across all drainage and age classes. Significant differences (p < 0.05) among the main six horizon codes are indicated with different letters. There are no thickness values for mineral soil because these results would reflect the thickness sampled, not the actual thickness of this horizon. Stdev is one standard deviation.

| Horizon Code | Bulk Density (g/cm³) | | | Carbon (%) | | | Nitrogen (%) | | | C:N | | |
|---|---|---|---|---|---|---|---|---|---|---|---|---|
| | mean | stdev | n | mean | stdev | n | mean | stdev | n | mean | stdev | n |
| live moss (L) | 0.022a | (0.018) | 138 | 41.7a | (3.8) | 145 | 0.84a | (0.25) | 145 | 53.8a | (16) | 141 |
| dead moss (D) | 0.039b | (0.026) | 540 | 42.6a | (3.8) | 538 | 0.77a | (0.27) | 537 | 62.1a | (23) | 541 |
| fibric (F) | 0.065c | (0.041) | 552 | 41.0a | (5.6) | 566 | 0.98a | (0.42) | 564 | 47.6a | (17) | 552 |
| mesic (M) | 0.149d | (0.077) | 634 | 38.2b | (6.8) | 650 | 1.42b | (0.54) | 651 | 30.6b | (13) | 634 |
| humic (H) | 0.215e | (0.096) | 160 | 32.1c | (6.6) | 164 | 1.53c | (0.44) | 164 | 22.2c | (6) | 160 |
| mineral (Min) | 0.731f | (0.380) | 584 | 6.5d | (6.2) | 674 | 0.34d | (0.32) | 673 | 18.0d | (7) | 603 |
| fibrous (D & F) | 0.052 | (0.037) | 1092 | 41.8 | (4.8) | 1104 | 0.88 | (0.37) | 1101 | 54.6 | (21) | 1101 |
| amorphous (M & H) | 0.162 | (0.085) | 794 | 36.9 | (7.2) | 814 | 1.44 | (0.52) | 815 | 28.9 | (12) | 813 |

*p value very close to 0.05 (thickness F vs M = 0.044). These values are so close to our threshold of 0.05 we would like to recognize that there is a chance that the bulk density values are not significantly different from each other.

**Table** 3. Thickness (cm) of the main horizon codes by soil drainage and age class. The mineral soil horizon was not included in this table because the way in which we sampled  mineral soil led to arbitrary thicknesses.  Significant differences (p < 0.05) for horizon codes among  drainage classes are indicated with different letters. Stdev is one standard deviation.

| Horizon | | Drainage | | | | | Age  | | |
|---|---|---|---|---|---|---|---|---|---|
| | | Well-drained | Moderately Well Drained | Somewhat Poorly Drained | Poorly Drained | Very Poorly Drained | New | Youn g | Mature |
| live moss (L) | mean | 2. (1.0) n=6a | 2. n=13 a | 2.1  n=a | 1.5  n=18a | 4. n=26b |  n=2ab | 2.6  n=43 a | 2.4  n=93b |
| | stdev | (1.0) | (0.8) | (1.1) | (0.7) | (2.1) | (-) | (2.) | (1.2) |
| | n | 6 | 11 | 75 | 18 | 26 | 2 | 42 | 92 |
| dead moss (D) | mean | 3.3a (1.6)  | 8.1a  n=20a | 7.5a  n=78a | 6.5a  n=21 | 38.8b  n=36b | 6.a (4.5) n=ab | 16.4b (19.7) n=45 a | 14.7b (30.2) n=99b |
| | stdev | (1.6) | (6.8) | (10.8) | (6.5) | (40.8) | (4.5) | (20.0) | (27.5) |
| | n | 6 | 19 | 77 | 21 | 34 | 17 | 42 | 98 |
| fibric (F)  | mean | 3.1a | 9.6bc | 7.9b | 13.7c | 40.2d | 6.4a | 19.8b | 14.0c |
| | stdev |  (3.0)  | 10.0abd  n=18 | 8.0b  n=12 | 13.6cd  n=4(10.9) | 39.1bd (38. n=27 | 6.6a (5. n=6 | 1.1b (31. n=41 |  (14.6) n=11 |
| | n | 11 | 18 | 121 | 45 | 26 | 65 | 39 | 117 |
| mesic (M) | mean | 2.8a | 13.3ab | 13.2ab | 14.4b | 57.2c | 6.3ab | 21.3a | 27.2b |

| | | | | | | | | | |
|---|---|---|---|---|---|---|---|---|---|
| mesics (M) | stdev | 2.8ª (1.3) n=5 | 12.4abc (16.7) n=(17.6) | 13.2b (37.9) n=113 | 15.2e (23(38.0) n=39 | 57.0d (53(21.8) n=34 | 6.5ab (4.1) n=54 (3.5) | 20.9a (32.6) n=53 (33.2) | 27.6b (51(53.4) n=101 (50.8) |
| | n | 5 | 15 | 112 | 39 | 33 | 53 | 50 | 101 |
| humic (H) | mean | none | 12.1ab | 5.6a | 7.4a | 20.2b | 4.3a | 13.1ab | 11.9b |
| humics (H) | stdev | none e / 10.0ab (14.0) n=9 -- | 6.2a (8.3) n=38(15.4) | (7.4b (3.4) n=13 | (3.4) | 20.7b (14.3) n=177) | 4.3a (3.2) n=24 | 13.4ab (12.7) n=19 | 12.3b (13.2) n=341) |
| | n | -- | 7 | 38 | 13 | 16 | 24 | 17 | 33 |
| fibrous (D & F) | mean | 4.5a | 14.8b | 11.3b | 14.8b | 58.5c | 7.1a | 27.0a | 22.9b |
| fibrous (D&F) | stdev | (4.5a (4.4) n=12) | (8.1) | 15.5b (8.9) n=22(9.9) | 11.6b (10.0) n=136 | 14.6b (11.3) n=522) | 59.8e (50.0) n=41(47.8) | 7.3a (6.4) n=733) | 26.7 (36.1) n=477) 23.5 (28.8) n=133(26.9) |
| | n | 12 | 21 | 135 | 51 | 40 | 73 | 54 | 132 |
| amorphous (M & H) | mean | 2.8a | 15.8a | 14.3a | 16.1a | 63.2b | 7.7a | 23.0a | 29.9b |
| amorphous (M&H) | stdev | 2.8a (1.3) n=5 | 15.8a (24.5) n=19(25.3) | 14.5a (38.3) n=1194) | 16.8a (22.3) n=41(21.0) | 63.6b (51.5) n=36 | 8.0 (5.3)a n=57(4.5) | 23.5a (33.2) n=585) | 30.5b (52.5) n=105(51.9) |
| | n | 5 | 18 | 118 | 41 | 35 | 56 | 56 | 105 |

Table 3.4. Physical and chemical properties of additional surface horizons. Number of observations, bulk density (g/cm³), C (%), N (%), C:N ratio, and thickness (cm) of non-main horizon codes. Values in parenthesis are standard deviations.

| Horizon | N | Bulk density (g/cm³) | Carbon (%) | Nitrogen (%) | C:N Ratio | Thickness (cm) |
|---|---|---|---|---|---|---|
| ash | 14 | 0.183 (0.155) | 38.0 (14.4) | 0.84 (0.34) | 49 (20) | 0.1 (-) |
| burned organics | 99 | 0.122 (0.142) | 38.6 (8.9) | 1.07 (0.32) | 99 (38) | 1. (1.0) |
| lichen | 31 | 0.034 (0.019) | 40.3 (5.9) | 0.76 (0.41) | 69 (37) | 4.1 (3.1) |
| litter | 16 | 0.044 (0.018) | 41.2 (3.1) | 1.55 (0.52) | 29 (10) | 1.6 (0.9) |

**Table 5.** Data columns found in megaAlaska_v11-2 for ScienceBase.csv. This datafile can be found at https://doi.org/10.5066/P960N1F9: Data Supporting Generalized models to estimate carbon and nitrogen stocks of organic layers in Interior Alaska.

| Column Name | Units | Column Description |
|---|---|---|
| sampleID | -- | The first four characters are based on the region and site. Then there is a space. Next the soil core number, followed by a period, and then the basal depth of the soil horizon. |
| depth | cm | Basal depth of the soil horizon |
| Hcode | -- | Horizon code as determined from Table 1 |
| Sample | -- | Qualitative description of the soil horizon |
| date | mm/dd/yy | Date sample was taken |
| thickness | cm | Thickness of the soil horizon |
| BDall | g/cm$^3$ | Bulk density, all soil |
| BDfine | g/cm$^3$ | Bulk density, fines (soil particles > 2 mm and roots > 1 cm diameter excluded) |
| HtAboveMin | cm | Height of each basal depth above the organic-mineral soil boundary |
| carbon | % | Carbon concentration |
| nitrogen | % | Nitrogen concentration |
| 13C | ‰ | Per mil (‰) value of delta $^{13}$C |
| 14C | ‰ | Per mil (‰) value of delta $^{14}$C for bulk soil sample |
| LOI | % | Loss-on-ignition value |
| volume_method | -- | Method used to sample soils volumetrically |
| region | -- | Region within Alaska where the site is located (Figure 1) |
| site | -- | Site where the core was taken |
| profile | -- | Soil profile, or core, number |
| drainage | -- | Soil drainage category (Figure 2) |
| standage | yrs | Age from last disturbance (fire or thaw) |
| ageclass | -- | N = newly burned (< 5 yrs), Y= young (5-50 yrs), M = mature (>50 yrs) |
| SurfaceVeg | -- | Types of vegetation found on the soil surface |
| SubbedBD | -- | If Y the bulk density is not a measured value. Instead an average value was used. |
| SubbedC | -- | If Y the carbon concentration is not a measured value. Instead an average value was used. |
| SubbedN | -- | If Y nitrogen concentration is not a measured value. Instead an average value was used. |
| GroupedHcode | -- | Horizon codes grouped into fewer categories |
| GroupedVeg | -- | Surface vegetation grouped into fewer categories |

**Table 6:** Data columns found in Site_GPS_coordinates_v2. This datafile can be found at https://doi.org/10.5066/P960N1F9: Data Supporting Generalized models to estimate carbon and nitrogen stocks of organic layers in Interior Alaska.

| Column Name | Description |
|---|---|
| Region | Region within Alaska where the site is located (Figure 1) |
| Region Code | Two letter code for the region |
| Site | Site where the core was taken |
| Profile | Which soil profiles are located at this location - all indicates general coordinates for all soil profiles |
| Latitude | Latitude in decimal degrees |
| Longitude | Longitude in decimal degrees |
| Datum | Datum of the coordinates |